# Micropropagation, Characterization, and Conservation of *Phytophthora cinnamomi*-Tolerant Holm Oak Mature Trees

Mª Teresa Martínez [1], Isabel Arrillaga [2], Ester Sales [3], María Amparo Pérez-Oliver [2], Mª del Carmen González-Mas [2] and Elena Corredoira [1],*

[1] Instituto de Investigaciones Agrobiológicas de Galicia (IIAG), Avda. Vigo s/n, 15705 Santiago de Compostela, Spain; temar@iiag.csic.es

[2] Departamento Biología Vegetal, Instituto Biotec/Med, Facultad de Farmacia, Universitat de València, Avda, Vicent Andrés Estellés s/n, 46100 Burjassot, Spain; isabel.arrillaga@uv.es (I.A.); mapeo5@uv.es (M.A.P.-O.); Carmen.Gonzalez.Mas@uv.es (M.d.C.G.-M.)

[3] Departamento Ciencias Agrarias y del Medio Natural, Instituto Universitario de Ciencias Ambientales (IUCA), Universidad de Zaragoza, Escuela Politécnica Superior, Ctra. Cuarte s/n, 22071 Huesca, Spain; esalesc@unizar.es

* Correspondence: elenac@iiag.csic.es

**Abstract:** Holm oak populations have deteriorated drastically due to oak decline syndrome. The first objective of the present study was to investigate the use of axillary budding and somatic embryogenesis (SE) to propagate asymptomatic holm oak genotypes identified in disease hotspots in Spain. Axillary budding was achieved in two out of six tolerant genotypes from the south-western region and in two out of four genotypes from the Mediterranean region. Rooting of shoots cultured on medium supplemented with 3 mg L$^{-1}$ of indole-3-acetic acid plus 0.1 mg L$^{-1}$ α-naphthalene acetic acid was achieved, with rates ranging from 8 to 36%. Shoot cultures remained viable after cold storage for 9–12 months; this procedure is therefore suitable for medium-term conservation of holm oak germplasm. SE was induced in two out of the three genotypes tested, by using nodes and shoot tips cultured in medium without plant growth regulators. In vitro cloned progenies of the tolerant genotypes PL-T2 and VA5 inhibited growth of *Phytophthora cinnamomi* mycelia when exposed to the oomycete in vitro. Significant differences in total phenol contents and in the expression profiles of genes regulating phenylpropanoid biosynthesis were observed between in vitro cultured shoots derived from tolerant trees and cultures established from control genotypes.

**Keywords:** axillary budding; disease-tolerant trees; dual cultures; gene expression; in vitro conservation; oak decline; *Quercus ilex*; phenols; *Phytophthora cinnamomi*; somatic embryogenesis

## 1. Introduction

Holm oak (*Quercus ilex* L.) is one of the most widely distributed tree species in the Mediterranean basin and is the dominant tree species in the dehesas. Dehesas are a good example of an agroforestry system of high natural and cultural value, where trees and shrubs, native grasses, crops, and livestock interact positively in managed areas, providing a large number of products with high economic value [1–3]. Since the 1990s, and particularly in recent years, dehesas have been seriously affected by oak decline disease or "la seca", which jeopardizes the long-term survival of this valuable ecosystem. The etiology of the disease is multiple and usually results from the interaction of several factors [4], such as aging and low natural regeneration of trees [5], the appearance of extreme climate events promoted by climate change including increased temperatures and altered precipitation regimes [6–8], and the emergence of new diseases and outbreaks of soil pathogens, mainly *Phytophthora cinnamomi* Rands (*Pc*) [9,10]. The incidence of holm oak decline is increasing and affecting more and more areas in the Mediterranean region [11]. To date, chemical control of the dispersion and infectivity of the oomycete, by applying calcium amendments

to the soil [12], injecting phosphates in the trunk [13], and by fumigating with plant extracts [14], have been relatively successful [15]. Resistant/tolerant trees can also be generated for reforesting areas affected by the disease. Theoretically, tolerance could be induced in conventional breeding programmes; however, no long-term tree improvement programmes have been carried out with oak species or other long-rotation hardwood species [16–18]. Investment in projects with an expected duration of more than 40 years, such as projects involving long-rotation hardwoods, is generally never attractive [17]. A more realistic approach would be to select mature pathogen-tolerant trees growing in disease hotspots and to use these for vegetative propagation, which has the advantage of capturing all of the genetic superiority without involving any gene segregation [18]. With this objective, various ongoing programs have identified and characterized several adult holm oak trees in disease hotspots. Exploiting these superior genotypes is difficult, however, since until now, holm oak has proven recalcitrant to conventional vegetative propagation.

The use of biotechnological approaches may be a valuable option regarding the propagation and conservation of selected genotypes [19], with axillary shoot proliferation being the most widely used method for vegetative propagation [20]. Unfortunately, in holm oak, only one report deals on micropropagation using explants derived from mature trees [21]. In this species, when adult plant material is used for axillary budding, it is usually difficult to stabilize shoot cultures in vitro, and rooting rates are generally low or null. By contrast, in other *Quercus* species, such as cork oak, pedunculate oak, and American oaks, the use of axillary budding has been extensively studied, as reviewed in [17]. On the other hand, somatic embryogenesis (SE) has been recognized as the most efficient system for regenerating woody plants, as well as a powerful tool for improving and conserving recalcitrant species, such as holm oak [22]. Somatic embryogenesis in holm oak has been more widely studied than axillary budding. Somatic embryos have been induced from holm oak immature zygotic embryos [23], male catkins [24], and teguments [25]. Recently, studies carried out by our group showed, for the first time, the possibility of inducing SE in apex explants and in leaves excised from adult tree-derived shoots established in vitro [26].

As holm oak forests and especially the dehesas are greatly altered, the conservation procedures for this species are challenging [27,28]. In vitro techniques undoubtedly play a role in ex situ conservation strategies, especially when the aim is to conserve specific genotypes or species with unorthodox seeds that lose their viability during storage. Axillary budding enables conservation of selected genotypes but is expensive and has the additional risk of contamination and somaclonal variation [29]. To prevent these risks, the period between each subculture event could be increased (thus reducing costs) by applying slow growth methods [30,31] or by completely inhibiting growth (cryobiotechnology) [29,32]. To date, very few studies have investigated preservation procedures in holm oak, such as cold storage to preserve established axillary shoots from juvenile plants [30], while in [33], the authors reported the cryopreservation of holm oak somatic embryos in a procedure that did not enable long-term preservation.

In vitro techniques, such as dual cultures, can also be applied to evaluate the pathogenicity of fungi and susceptibility of host plant genotypes as a complementary method [34]. Furthermore, this technique was used to study the potential of chemical elicitation treatments to induce defense responses to *P. cinnamomi* on holm oak embryogenic lines [35]. Plants have developed a multi-layered defense network to detect invading pathogens and stop them before they cause extensive damage. The phenylpropanoid pathway is important in a plant's defense mechanisms and provides structural and chemical barriers for resistance to pathogen infection [36]. In fact, during pathogen attack, phenylpropanoid pathway genes were found to be overexpressed, resulting in increased enzymatic activities and accumulation of various phenolic compounds [37], and the presence of phenolic compounds in root exudates has been associated with the tolerance prevention strategy against Black Shank Disease in tobacco [38].

In order to alleviate the ecological and economic impact of oak decline on holm oak populations in the Iberian Peninsula, the main objectives of the present study were:

(1) regeneration by axillary shoot proliferation of mature trees growing in oak decline hotspots and selected for their tolerance to *Phytophthora*; (2) induction of SE from explants excised from in vitro shoot cultures; (3) medium-term preservation, by cold storage, of in vitro shoot cultures established from valuable trees; and (4) characterization of the tolerant phenotype in micropropagated trees by dual cultures and by determination of the total phenols content and gene expression profile of some of the genes involved in phenylpropanoid biosynthesis.

## 2. Materials and Methods

### 2.1. Plant and Oomycete Material

The source of plant material consisted of branches collected in January and March 2017 from *Quercus ilex* trees aged 30–100 years, which were growing in three locations of Spain: two intensely affected by oak decline, La Vallivana in Castellón and Plasencia in Cáceres, and the other without affected trees in La Hunde, Valencia. Cáceres is a province located in the south-west of Spain, while Castellón and Valencia are provinces from the Mediterranean area of Spain; therefore, we used genotypes from two provenances. A total of 10 holm oak genotypes without decline symptoms were sampled: six in Plasencia, called PL-T1, PL-T2, PL-T3, PL-T4, PL-T5, and PL-T6; two in La Vallivana, named VA5 and VA11; and another two in La Hunde, named H1 and H5 (see Table S1 for localization details). In all genotypes, branches were collected from the crown (C), except for VA11, from which shoots were also collected from the basal part of the tree (Bs). The presence of *P. cinnamomi* was confirmed in the soils in which the trees grew, except for the VA11 individual. In some experiments, axillary shoot cultures previously established from 4 adult genotypes (E00, E2, Q3, and Q10), as described [21], were also used. These trees were selected for fruit production in an area not affected by oak decline.

*Phytophthora cinnamomi* strain 1630 was kindly provided by Dr. Paloma Abad (group Phytopathogenic fungi, Instituto Agroforestal Mediterráneo-Universidad Politécnica de Valencia, Spain) and was maintained in PDA medium (Potato Dextrose Agar, Pronadisa, Spain) by subculturing mycelium pieces of $0.5\ cm^2$ to fresh medium every 15 days.

### 2.2. In Vitro Culture Establishment

Branch segments (25–30 cm long and 2–4 cm thick) were treated with a fungicide (Cupravit, Bayer, Leverkusen, Germany; $3.5\ g\ L^{-1}$) for 1 h. Following this treatment, branches were dried at room temperature for 24 h and then placed on a vertical position on a plastic tray that was not sterile with moistened perlite that was not sterile, which was put on the phytotron, with 90% relative humidity and under a 16 h photoperiod ($90–100\ \mu mol\ m^{-2}\ s^{-1}$ provided by cool-white fluorescent lamps; $25 \pm 1\ °C$ light/$20 \pm 1\ °C$ in darkness). After approximately 2–3 weeks on the phytotron, the development of epicormic shoots was observed (Figure 1A,B). Flushing capacity was evaluated by recording the segments that produced newly flushed shoots, the number of shoots per branch, and the mean shoot length (mm). Defoliated shoots were superficially sterilized with a 0.3% (*w/v*) sodium hypochlorite (Millipore®, Merck, Kenilworth, NJ, USA) solution containing 2–3 drops of Tween 80® for 2.5 min. The hypochlorite was eliminated with two 10 min washes of sterile water.

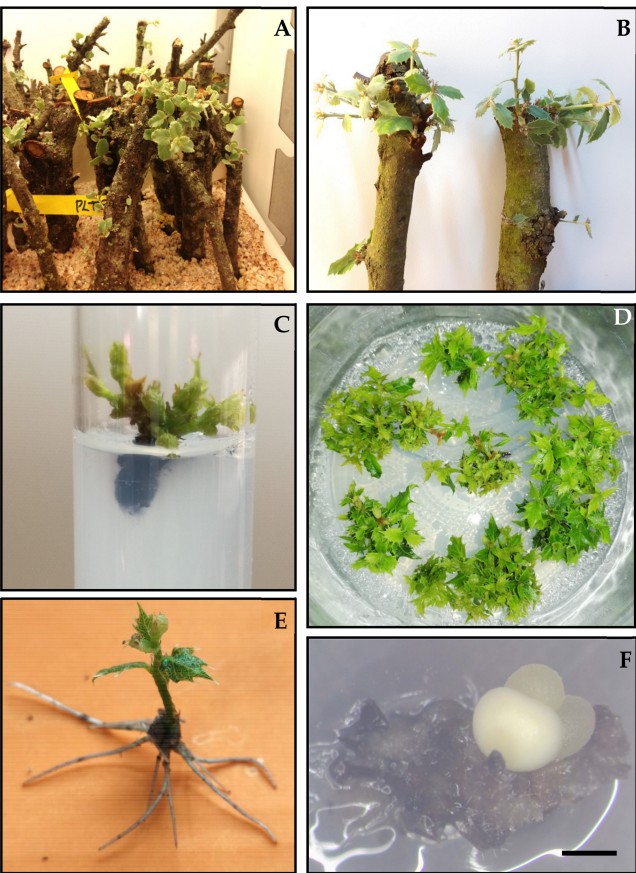

**Figure 1.** Micropropagation of adult holm oak trees selected for their ability to grow in disease hotspots. (**A**,**B**) Forced flushing of branch segments from trees from the south-west area (**A**) and Mediterranean area (**B**) in a climate chamber. (**C**) Shoot development from a nodal explant collected from a forced branch shoot 8 weeks after initial culture. Tube diameter: 30 mm. (**D**) Appearance of shoot cultures after 6 weeks on proliferation medium. Jar diameter: 90 mm. (**E**) Rooted shoot of clone VA5 after culture for 48 h on the rooting medium supplemented with IBA 25 mg L$^{-1}$ and 6 weeks on rooting medium without plant growth regulators. (**F**) Induction of somatic embryos from an apex explant of VA11-C clone cultured on induction medium without plant growth regulators. Bar: 1 mm.

Following surface sterilization, nodal segments with 1–2 axillary buds and approximately 0.5 cm long were cut and used as initial explants. Explants were put vertically on tubes (30 × 150 mm) containing 20 mL of establishment medium, which consisted of Woody Plant medium (WPM) (Duchefa Biochemie, Haarlem, The Netherlands) [39] supplemented with 30 g L$^{-1}$ sucrose, 6 g L$^{-1}$ Plant Propagation Agar (PPA; Condalab, Madrid, Spain), 80 mg L$^{-1}$ ascorbic acid, and 0.5 mg L$^{-1}$ benzyladenine (BA). The pH of this medium was adjusted to 5.6–5.7 before autoclaving at 115 °C for 20 min. Ascorbic acid was sterilized by filtration and added to the autoclaved medium. To avoid browning, after 24 h, the explants were changed to the opposite position in the same tube and transferred every 2 weeks to fresh medium of the same composition until reaching 8 weeks of culture. After this time, cultures were checked to determine the contamination rate and shoot responsiveness (percentage of explants with new shoots ≥ 5 mm).

Unless otherwise specified, cultures were kept in growth chambers at 20 °C dark/25 °C light. Illumination was provided by white cold fluorescent tubes (50–60 µmol m$^{-2}$ s$^{-1}$) for 16 h (i.e., standard culture conditions).

### 2.3. Shoot Stabilization and Proliferation

After 8 weeks of culture, new flushed shoots developed from initial explants and higher than 0.5 cm were isolated and cultured on jars containing 70 mL of proliferation

medium. This medium is based on our previous studies [21] and consisted of WPM supplemented with 30 g L$^{-1}$ sucrose, 8 g L$^{-1}$ Sigma agar (A-1296; Sigma-Aldrich, St. Louis, MO, USA), 20 µM filter-sterilized silver thiosulphate (STS) (Millipore, Merck, Darmstadt, Germany), and BA. The concentration of BA was periodically reduced following a 6-week cycle: 0.1 mg L$^{-1}$ BA for the first 2 weeks, 0.05 mg L$^{-1}$ BA for the next 2 weeks, and 0.05 mg L$^{-1}$ BA for the last 2 weeks. Shoot cultures were maintained by repeating these conditions until stabilization of the proliferation rates was attained. The effect of the genotype on the shoot proliferation capacity of holm oak adult material was then studied, using shoots of clones PL-T2, VA5, VA11-Bs, and VA11-C. The following parameters were evaluated at the end of the 6-week proliferation cycle: responsive explants (i.e., the percentage of initial explants that formed new shoots), total mean number of shoots produced per explant, mean number of shoots 0.5–1.0 cm per explant, mean number of shoots > 1.0 cm per explant, and mean length of the longest shoot among the responsive explants. In each genotype, five replicate jars (each containing seven explants) were evaluated, and the experiment was repeated at least twice (i.e., 70 explants in total).

## 2.4. Shoot Rooting

To evaluate their rooting capacity, shoots (1.0–1.5 cm long) isolated from 6-week-old cultures of clones VA5, VA11-Bs, and VA11-C were employed. In the first experiment, shoots were cultured for 15 days on Murashige and Skoog medium (MS) (Duchefa Biochemie) [40] with half-strength macronutrients ($\frac{1}{2}$ MS) containing 30 g L$^{-1}$ sucrose, 6 g L$^{-1}$ PPA, 3 mg L$^{-1}$ indole−3−butyric acid (IBA) (Duchefa Biochemie), and 0.1 mg L$^{-1}$ α-naphthalene acetic acid (NAA) (Duchefa Biochemie). After 15 days on rooting medium, shoots were transferred to the same medium without auxins and supplemented with 20 µM STS.

In a second experiment, shoots were cultured in Gresshoff and Doy medium (GD) (Duchefa Biochemie) [41] with macronutrients reduced to one-third strength ($^1/_3$ GD), supplemented with 30 g L$^{-1}$ sucrose, 6 g L$^{-1}$ PPA, and 25 mg L$^{-1}$ IBA. After 24 or 48 h on rooting medium, shoots were transferred to the same medium without IBA and supplemented with 20 µM STS. In a third experiment, shoots were cultured in $^1/_3$ GD medium supplemented with 3 mg L$^{-1}$ IBA and 200 mg L$^{-1}$ phloroglucinol for 6 weeks.

In each rooting experiment, 25 shoots were used per clone and treatment, and each experiment was repeated at least twice (i.e., 50 explants in total). The percentage of rooted shoots, the mean number of roots per rooted shoot, and the length of the longest root (mm) from each shoot was determined after 6 weeks.

## 2.5. Cold Storage of Shoot Cultures

In these experiments, we used axillary shoot cultures previously established from holm oak mature trees selected for fruit production (Q3, Q10, and E00) [21] and the clones established in the present report (PL-T2, VA5, VA11Bs, and VA-11-C) to develop a simple and efficient procedure for in vitro conservation of valuable holm oak genotypes. Clumps with 2–3 axillary buds isolated from 6-week-old shoot cultures were maintained for 2 weeks on standard conditions and then stored in a cold chamber (340 L Sanyo Medicool Cabinets) at 4 ± 1 °C under dim light conditions (16 h photoperiod, 8–10 µmol m$^{-2}$ s$^{-1}$) for 0 (control), 6, 9, and 12 months. At the end of each storage period, cultures were immediately transferred to fresh proliferation medium and placed under the standard proliferation conditions described above. After 6 weeks of culture, the effect of cold storage on shoot proliferation was evaluated by recording the following parameters: recovery rate (defined as the percentage of explants that developed new shoots), total mean number of shoots produced per explant, mean number of shoots 0.5–1.0 cm per explant, mean number of shoots > 1.0 cm per explant, and the mean length of the longest shoot. For each genotype and cold storage period, five replicated jars (each containing seven explants) were evaluated (i.e., 35 explants in total).

*2.6. Somatic Embryogenesis*

Axillary shoot cultures of four clones PL-T2, VA5, VA11-Bs, and VA11-C were used as the source of explants for inducing somatic embryo formation. Explants consisted of (1) the most apical expanding leaf excised from the first node in the apical region, (2) shoot apices (2–2.5 mm long, comprising the apical meristem and several pairs of leaf primordia), and (3) the first node in the apical region, which were isolated from shoots actively growing from VA11-Bs and VA11-C clones. Explants were cultured following a two-step procedure: initially in the dark for 2, 4, or 8 weeks; in induction medium (M1) consisting of MS salts, 500 mg L$^{-1}$ casein hydrolysate (HC), 6 g L$^{-1}$ PPA, and 30 g L$^{-1}$ sucrose, and supplemented with 0.5 mg L$^{-1}$ BA combined with 4 mg L$^{-1}$ of either indole-3-acetic acid (IAA) (Duchefa Biochemie) or NAA; or supplemented with 3 mg L$^{-1}$ IBA in combination with 0.1 mg L$^{-1}$ NAA. In the second step, explants were transferred to the same basal medium without plant growth regulators (expression medium, M2) for at least 24 weeks (i.e., in total, explants were cultured for 36 weeks). In a second experiment, shoot apex and node explants of four clones were cultured on M2, that is, without plant growth regulators (PGRs), for 36 weeks. In total, 100 explants (10 Petri dishes with 10 explants per dish) were used for each explant type, clone, and treatment. Embryogenic ability was defined as the presence of somatic embryos and/or nodular embryogenic structures (NSs) in the initial explants. The embryogenic response was examined under a stereomicroscope (Olympus SC1.00, Tokyo, Japan) and photographed with an Olympus DP10 digital camera (Japan).

*2.7. Dual Cultures*

In vitro grown shoots, routinely maintained in glass jars with proliferation medium as previously described in Section 2.3, of two uncharacterized (E00 and E2) and two tolerant (PL-T2 and VA5) *Q. ilex* genotypes were employed in two experiments. Dual cultures were established with either three pairs of leaves or three shoots from one of the uncharacterized genotype controls and from a tolerant genotype, which were combined in Petri dishes (14 cm Ø and 2 cm high) containing Schenk and Hildebrand medium (SH) (Duchefa Biochemie) [42], and confronted with a section (0.5 cm side) of *Pc* mycelium in potato-dextrose-agar medium, as described by Morcilllo et al. [35]. Each genotype was tested in 4 plates. Hyphae length towards each shoot was determined after 3, 6, 10, and 17 days, while in the experiment performed with leaf explants, the oomycete mycelium growth was recorded after 2, 4, and 7 days.

*2.8. Total Phenols Determinations*

In vitro grown shoots from E00, E2, PL-T2, and VA5 *Q. ilex* genotypes were sampled among those challenged with *Pc* in dual cultures, and shoots growing in non-infected plates were sampled as controls. These shoots were dried at 60 °C to constant weight (3 days) and, after dry weight (DW) determinations, samples were manually homogenized in a mortar and stored at 4 °C until analysis. For each genotype and treatment, total phenols were determined twice in duplicated samples, following a modified Folin–Ciocalteu method, as described in [43]. Absorbance at 760 nm was measured using a UV-visible spectrophotometer (Biospectrometer, Eppendorf, Hamburg, Germany). Results were expressed in gallic acid equivalents (GAE); that is, µg gallic acid/g DW, using a gallic acid standard curve (40–340 µg/g).

*2.9. Gene Expression Analyses*

We investigated by qPCR the expression of four genes coding for key enzymes involved in the phenolic compounds synthesis: *chorismate synthase* (*CS*), *phenylalanine ammonia-lyase* (*PAL*), *NADPH-dependent cinnamyl alcohol dehydrogenase* (*CAD*1), and *chalcone synthase* (*ChS*), using primers (Table 1) designed for *Quercus suber* [44]. An *elongation factor* (*EF*) was used as the reference gene, with the primers designed in [45].

**Table 1.** Primers used for qPCR assays to amplify the chorismate synthase (*CS*), phenylalanine ammonia-lyase (*PAL*), NADPH-dependent cinnamyl alcohol dehydrogenase (*CAD*1), and chalcone synthase (*ChS*) genes. Elongation factor (EF) was used as reference gene.

| Gene | Forward Primer | Reverse Primer |
|------|----------------|----------------|
| *CAD* | CAGATGATAAGCCATTTGCG | AGGAACTTCAGGGTGCTAC |
| *ChS* | TGAGATCACAGCAGTTAC | CAAGTTGAAACAGTGGAC |
| *CS* | TGGATTGATTGGAAACAGATTAC | CAAGGAAGCAGCACACAG |
| *PAL* | ATTAGCAGGGATTGATGG | CAAGTGGTCTGTAAATTCG |
| *EF* | TTGTGCCGTCCTCATTATTGACT | TCACGGGTCTGACCATCCTT |

Total RNA was isolated, using the Plant and Fungi RNA isolation kit (Norgen Biotek, Thorold, Canada), from leaves sampled in 30-day-old in vitro grown E00, PL-T2, and VA5 *Q. ilex* shoots, cultured in jars containing WPM supplemented with BA and 20 μM STS, as described in Section 2.3. RNA samples were also prepared from leaves sampled in 17-day-old shoots of these genotypes confronted with *Pc* in dual cultures. RNA was purified with DNase (Takara Bio Inc., Kusatsu, Japan) and quantified with a NanoDrop™ One (Thermo Fisher Scientific, USA). Real-time amplifications were performed in triplicate using a Step One Plus thermocycler (Applied Biosystems, USA), with 1 μL of cDNA synthetized with a reverse transcriptase kit (Takara Bio Inc.). The reaction mix for real-time PCR (final volume 20 μL) was prepared with the SYBR® Premix Ex Taq™ (Tli RNaseH Plus, Takara Bio Inc.), following the manufacturer's specifications, and with 0.3 μM of each primer. The amplification program consisted of 40 cycles of 3 s at 95 °C and 30 s at 60 °C. The primers' efficiency was checked using serial dilutions (1:4) of cDNA. Gene expression was estimated as $2^{-\Delta CT}$ [46].

*2.10. Statistical Analysis*

Statistical analysis was performed with SPSS for Windows (version 25.0, Chicago, IL, USA). Data from axillary budding and somatic embryogenesis were statistically analyzed by analysis of variance (ANOVA) one-way factorial (ANOVA I) or two-way factorial (ANOVA II). Percentage data were subjected to arcsine transformation before the analysis and non-transformed data are presented in the tables. Data from *P. cinnamomi* mycelium growth, total phenols content, and gene expression were also subjected to ANOVA. When values did not adjust to normal distribution, a non-parametric Kruskal–Wallis test was used.

**3. Results**

*3.1. Micropropagation by Axillary Budding*

3.1.1. Establishment, Stabilization, and Proliferation of Axillary Shoot Cultures

Forced flushing on branch segments (Figure 1A,B) was achieved in all genotypes except for PL-T6, probably because this tree could actually be affected by oak decline (Table 2). The new shoots emerged after 2–4 weeks of controlled flushing in a climate chamber. However, large differences were observed in the flushing capacity of the branch segments of trees growing in different areas. The mean sprouting rate was much higher in branches sampled in trees from the Mediterranean provenance (mean 85%, range 68.2–100%) than in those from the south-west provenance (mean 39%, range 0–70.9%). Likewise, the mean length of the new formed shoots was much higher in the branch segments from La Vallivana and La Hunde (Mediterranean area) than in those from Plasencia (south-west area), ranging between 16.2–43.7 and 12.5–15.4 mm, respectively (Table 2).

**Table 2.** Response to sprouting and in vitro establishment of adult oak genotypes from two Spanish regions, Mediterranean and South-west areas.

| Clones | Flushing Capacity of Branch Segments (4 w) [1] | | | | Response to In Vitro Establishment (8 w) [2] | | | |
|---|---|---|---|---|---|---|---|---|
| | Branch Segment Number | Branch with Sprouting (%) | Total Number Shoots | Length of Shoots (mm) | Number of Initial Explants | Contamination Rate (%) | Response Rate (%) [3] | Shoot Establishment |
| Mediterranean | | | | | | | | |
| H1 | 30 | 86.7 | 48 | $16.2 \pm 5.8$ | 66 | 18.3 | 21.8 | No |
| H5 | 22 | 68.2 | 58 | $20.1 \pm 5.8$ | 94 | 27.7 | 26.5 | No |
| VA5 | 32 | 96.9 | 109 | $30.1 \pm 10.5$ | 206 | 2.4 | 30.9 | Yes |
| VA11-Bs [4] | 21 | 100.0 | 131 | $43.7 \pm 20.2$ | 225 | 15.6 | 15.8 | Yes |
| VA11-C [4] | 12 | 75.0 | 33 | $18.4 \pm 7.2$ | 51 | 2.0 | 42.0 | Yes |
| South-West | | | | | | | | |
| PL-T1 | 26 | 30.8 | 36 | $14.7 \pm 5.2$ | 41 | 61.0 | 100.0 | Yes |
| PL-T2 | 31 | 70.9 | 57 | $15.4 \pm 7.7$ | 96 | 52.1 | 15.2 | Yes |
| PL-T3 | 23 | 65.2 | 40 | $13.9 \pm 6.4$ | 59 | 25.4 | 11.4 | No |
| PL-T4 | 32 | 31.3 | 19 | $12.9 \pm 4.8$ | 26 | 34.6 | 11.8 | No |
| PL-T5 | 12 | 33.3 | 6 | $12.5 \pm 3.7$ | 6 | 33.3 | 0.0 | - |
| PL-T6 | 32 | 0.0 | 0 | - | - | - | - | - |

[1] Evaluated after 4 weeks in growth chamber; [2] evaluated after 8 weeks in growth chamber; [3] in vitro explants with sprouting buds after 8 weeks of culture; [4] branches from the basal (Bs) or crown (C) part of the VA11 tree.

Regarding the in vitro establishment response, a high incidence of contamination occurred in the initial cultures derived from the Plasencia genotypes (25.4% to 61%) relative to the Mediterranean genotypes (2% to 27.7%) (Table 2). In addition, after inoculation of new formed shoots in culture medium, browning was also observed in all the studied genotypes. The addition of ascorbic acid to the culture medium and transfer of explants to the opposite side of the test tube within the first 24 h did not prevent the negative effects of browning, such as limited shoot growth, necrosis, and even death of the explants. The formation of new shoots in vitro is independent of the flushing capacity of branches of each genotype. All the tested genotypes except PL-T5 formed buds (≥5 mm) in vitro after 8 weeks of culture and, with the exception of genotype PL-T1, response rates were higher in the Mediterranean genotypes than in those from the south-west area (Table 2). No differences were observed in the response rates of VA11-C and VA11-Bs clones derived from the same genotype, in spite of their different ontogenic origin.

After 8 weeks of culture, shoots longer than 5 mm, which had developed from original explants (Figure 1C), were isolated and successively cultured on proliferation medium to stabilize shoot cultures. Stable proliferating cultures were obtained in two out of the four genotypes established from Plasencia (PL-T1 and PL-T2) and in two out of the four genotypes from the Mediterranean locations (VA5 and VA11-Bs and C). The duration of the stabilization phase, which is required for uniform growth of shoots, was highly variable, ranging from 4 to 12 months depending on the genotype.

Following stabilization, shoot multiplication on the proliferation medium was successively repeated in four clones until a sufficient number of shoots was produced for evaluating the proliferation capacity (Table 3). With the exception of the percentage of responsive explants, all parameters evaluated were significantly affected ($p = 0.001$) by the genotype and the ontogenic origin in the case of genotype VA11. Satisfactory proliferation rates were obtained, especially with genotypes VA5 and VA11. The VA5 genotype yielded the greatest number of total shoots per explant, and the PL-T2 genotype yielded the lowest number (Table 3). The shoot length ranged between 13.3 and 22.4 cm, with the VA5 genotype again showing better results (Table 3). In plant material from the VA11 tree, proliferation rates were affected by the ontogenetic age of the established material, since VA11-C cultures, established from the crown material, yielded a greater total number of shoots than those determined in VA11-Bs cultures that established sprouted branches from the base of the tree. However, VA11-Bs proliferating cultures produced more shoots > 1 cm (Table 3) than VA11-C cultures. Despite the obvious differences between genotypes, the shoot proliferation rates were sufficient to provide a continuous source of shoots (Figure 1D) to perform further experiments of adventitious rooting, SE induction, and dual cultures.

**Table 3.** Effect of genotype and/or ontogenic origin on in vitro shoot multiplication of adult trees of *Q. ilex*.

| Clone | Responsive Explants (%) | Total Shoots (N°) | Shoots 0.5–1.0 (N°) | Shoots ≥ 1 cm (N°) | Longest Shoot Length (mm) |
|---|---|---|---|---|---|
| VA11-Bs | 100 ± 0.0 | 7.82 ± 0.38 | 2.53 ± 0.27 | 5.29 ± 0.21 | 19.33 ± 0.48 |
| VA11-C | 100 ± 0.0 | 8.30 ± 0.17 | 4.96 ± 0.15 | 3.34 ± 0.12 | 15.57 ± 0.39 |
| VA5 | 100 ± 0.0 | 17.76 ± 0.56 | 10.33 ± 0.52 | 7.43 ± 0.31 | 22.40 ± 0.83 |
| PL-T2 | 100 ± 0.0 | 5.34 ± 0.17 | 1.96 ± 0.18 | 3.38 ± 0.16 | 13.30 ± 0.52 |
| ANOVA I | 1.0 ns | 0.001 *** | 0.001 *** | 0.001 *** | 0.001 *** |

Each value is the mean ± standard error of ten replicate jars with seven explants per jar. ANOVA I significances are shown for each parameter. ns: not significant; *** significant differences at 99.9% ($p \leq 0.001$).

### 3.1.2. Rooting of Axillary Shoots

In the first experiment, the rooting ability of shoot cultures of clones VA5, VA11-Bs, and VA11-C was evaluated after application of 3 mg L$^{-1}$ IBA and 0.1 mgL$^{-1}$ NAA for 15 days (Table 4). The rooting rate was significantly influenced by the genotype ($p = 0.012$) and was highest in VA5 (36%). In VA11 shoot cultures, rooting rates were higher on

shoot cultures derived from basal material (VA11-Bs) than on those derived from crown material (VA11-C). The number of roots per shoot varied from 1 to 4.4 and was significantly ($p = 0.004$) affected by the genotype and the ontogenic origin (Table 4). Genotype also significantly influenced ($p = 0.028$) root length, and genotypes VA5 and VA11-C produced the longest roots (47.5 and 40.0 mm, respectively).

**Table 4.** Effect of 3 mg $L^{-1}$ IBA plus 0.1 mg $L^{-1}$ NAA added into the rooting medium on rooting rates of clones VA5, VA11-Bs, and VA11-C.

| Clone | Rooting (%) | Root Number | Root Length (mm) |
|---|---|---|---|
| VA5 | $36.0 \pm 9.3$ | $4.4 \pm 0.7$ | $47.5 \pm 5.7$ |
| VA11-Bs | $12.0 \pm 3.1$ | $1.2 \pm 0.1$ | $17.3 \pm 2.1$ |
| VA11-C | $8.0 \pm 3.1$ | $1.0 \pm 0.0$ | $40.0 \pm 7.8$ |
| ANOVA I | 0.012 * | 0.004 ** | 0.028 * |

Each value is the mean $\pm$ standard error of 10 replicate jars with 5 explants per jar. ANOVA I significances are shown for each parameter. * significant differences at 95% ($p \leq 0.05$); ** significant differences at 99% ($p \leq 0.01$).

In the second experiment, the rooting capacity of the same clones was tested after application of 25 mg $L^{-1}$ IBA for 24 or 48 h (see the Table S2). Rooting rates were significantly affected by genotype ($p = 0.001$) and the interaction ($p = 0.035$) between genotype and duration of exposure to IBA (see the Table S2). Rooting was only achieved in clone VA5, and higher rates (68.3%) were obtained after exposure to 25 mg $L^{-1}$ IBA for 48 h (Figure 1E). In this genotype, rooting rates were higher than those obtained with the 3 mg $L^{-1}$ IBA treatment (Table 4).

Finally, investigation of the effect of culturing shoots on 3 mg $L^{-1}$ IBA plus 200 mg $L^{-1}$ phloroglucinol for 6 weeks showed that the latter compound had a deleterious effect on the in vitro rooting capacity of holm oak shoots (data not shown).

### 3.1.3. Medium-Term Conservation of Axillary Shoots

Axillary shoots of seven clones established from adult holm oak trees were stored at 4 °C to evaluate medium-term in vitro conservation. Recovery rates of 68.6–100% were obtained after cold storage for 12 months in six of the seven clones, but the PL-T2 clone could not be stored for more than 9 months (Table 5). The total number of recovered shoots was significantly influenced by the duration of cold storage in all clones except in Q3. In general, the number of shoots decreased as the duration of cold storage increased (Table 5), but it was sufficient for recovery of normal growth of shoot cultures after culture under standard conditions. Regarding the total number of shoots of length 0.5–1 cm or $\geq$1 cm, large differences were observed between the 7 clones evaluated.

**Table 5.** Effect of cold storage period on recovery and proliferation capacity of shoot cultures derived from 7 adult *Q. ilex* clones.

| CLONE | Storage Period (Months) | Recovery (%) [1] | Total Shoots (N°) | Shoots 0.5–1.0 cm (N°) | Shoots $\geq$1 cm (N°) | Longest Shoot Length (mm) |
|---|---|---|---|---|---|---|
| | 0 | $100 \pm 0.0$ | $3.2 \pm 0.2$ | $1.3 \pm 0.1$ | $1.9 \pm 0.2$ | $16.2 \pm 0.5$ |
| | 6 | $100 \pm 0.0$ | $4.8 \pm 0.3$ | $1.8 \pm 0.2$ | $3.0 \pm 0.2$ | $14.7 \pm 1.4$ |
| Q10 | 9 | $97.1 \pm 2.5$ | $4.3 \pm 0.3$ | $1.5 \pm 0.2$ | $2.8 \pm 0.2$ | $12.8 \pm 1.1$ |
| | 12 | $100 \pm 0.0$ | $3.7 \pm 0.2$ | $2.0 \pm 0.2$ | $1.7 \pm 0.2$ | $15.7 \pm 0.6$ |
| ANOVA I | | 0.418 ns | 0.008 ** | 0.045 * | 0.003 ** | 0.81 ns |
| | 0 | $100 \pm 0.0$ | $2.7 \pm 0.1$ | $1.7 \pm 0.2$ | $1.0 \pm 0.1$ | $16.2 \pm 0.5$ |
| | 6 | $78.6 \pm 4.0$ | $2.5 \pm 0.4$ | $1.4 \pm 0.2$ | $1.1 \pm 0.2$ | $14.5 \pm 0.3$ |
| Q3 | 9 | $64.3 \pm 6.3$ | $2.8 \pm 0.1$ | $1.6 \pm 0.2$ | $1.2 \pm 0.1$ | $11.2 \pm 0.5$ |
| | 12 | $100 \pm 0.0$ | $2.3 \pm 0.3$ | $1.2 \pm 0.2$ | $1.1 \pm 0.2$ | $11.2 \pm 2.4$ |
| ANOVA I | | 0.001 *** | 0.523 ns | 0.386 ns | 0.818 ns | 0.62 ns |

**Table 5.** *Cont.*

| CLONE | Storage Period (Months) | Recovery (%) [1] | Total Shoots (N°) | Shoots 0.5–1.0 cm (N°) | Shoots ≥1 cm (N°) | Longest Shoot Length (mm) |
|---|---|---|---|---|---|---|
| **E00** | 0 | 100 ± 0.0 | 4.4 ± 0.3 | 1.8 ± 0.1 | 2.6 ± 0.1 | 23.3 ± 1.0 |
| | 6 | 100 ± 0.0 | 3.1 ± 0.1 | 1.5 ± 0.1 | 1.6 ± 0.2 | 15.5 ± 0.2 |
| | 9 | 71.4 ± 13.3 | 2.5 ± 0.5 | 1.4 ± 0.1 | 1.1 ± 0.4 | 12.7 ± 2.9 |
| | 12 | 68.6 ± 8.4 | 1.8 ± 0.1 | 1.3 ± 0.1 | 0.5 ± 0.1 | 10.3 ± 0.9 |
| **ANOVA I** | | 0.012 * | 0.001 *** | 0.082 ns | 0.001 *** | 0.001 *** |
| **PL-T2** | 0 | 100 ± 0.0 | 5.5 ± 0.3 | 2.1 ± 0.3 | 3.4 ± 0.3 | 13.4 ± 0.9 |
| | 6 | 84.0 ± 3.4 | 4.0 ± 0.1 | 1.7 ± 0.1 | 2.3 ± 0.1 | 10.0 ± 0.8 |
| | 9 | 54.3 ± 15.8 | 1.9 ± 0.2 | 1.4 ± 0.3 | 0.5 ± 0.1 | 7.4 ± 0.9 |
| | 12 | - | - | - | - | - |
| **ANOVA I** | | 0.018 * | 0.001 *** | 0.207 ns | 0.001 *** | 0.006 ** |
| **VA11-C** | 0 | 100 ± 0.0 | 8.4 ± 0.2 | 4.9 ± 0.1 | 3.5 ± 0.1 | 16.4 ± 0.3 |
| | 6 | 100 ± 0.0 | 9.2 ± 0.5 | 5.8 ± 0.6 | 3.4 ± 0.2 | 19.7 ± 0.6 |
| | 9 | 100 ± 0.0 | 5.8 ± 0.3 | 3.1 ± 0.3 | 2.7 ± 0.2 | 18.4 ± 0.7 |
| | 12 | 100 ± 0.0 | 6.7 ± 0.2 | 4.0 ± 0.2 | 2.7 ± 0.2 | 18.7 ± 0.8 |
| **ANOVA I** | | 1.0 ns | 0.001 *** | 0.002 ** | 0.007 ** | 0.026 * |
| **VA11-Bs** | 0 | 100 ± 0.0 | 8.3 ± 0.4 | 2.9 ± 0.3 | 5.4 ± 0.1 | 19.3 ± 0.28 |
| | 6 | 100 ± 0.0 | 9.6 ± 0.5 | 5.7 ± 0.4 | 3.9 ± 0.2 | 19.7 ± 0.6 |
| | 9 | 100 ± 0.0 | 7.1 ± 0.2 | 4.4 ± 0.1 | 2.7 ± 0.2 | 17.9 ± 0.6 |
| | 12 | 100 ± 0.0 | 6.0 ± 0.5 | 3.1 ± 0.3 | 2.9 ± 0.3 | 19.2 ± 0.9 |
| **ANOVA I** | | 1.0 ns | 0.001 *** | 0.001 *** | 0.001 *** | 0.378 ns |
| **VA5** | 0 | 100 ± 0.0 | 18.5 ± 0.6 | 10.5 ± 0.6 | 8.0 ± 0.3 | 20.2 ± 0.6 |
| | 6 | 100 ± 0.0 | 15.3 ± 0.8 | 9.2 ± 0.3 | 6.1 ± 0.6 | 24.3 ± 0.9 |
| | 9 | 100 ± 0.0 | 10.2 ± 1.4 | 3.4 ± 0.6 | 6.8 ± 0.9 | 23.5 ± 0.7 |
| | 12 | 97.1 ± 2.5 | 5.2 ± 0.8 | 4.6 ± 0.7 | 0.6 ± 0.2 | 10.3 ± 0.7 |
| **ANOVA I** | | 0.418 ns | 0.001 *** | 0.001 *** | 0.001 *** | 0.001 *** |

Each value is the mean ± standard error of ten replicate jars with seven explants per jar. [1] Recovery rate defined as percentage of shoots showing growth after storage for 0, 6, 9 or 12 months at 4 °C and subsequent culture under standard conditions. ANOVA I significances are shown for each parameter. ns: not significant; * significant differences at 95% ($p \leq 0.05$); ** significant differences at 99% ($p \leq 0.01$); *** significant differences at 99.9% ($p \leq 0.001$).

Usually, the length of the longest shoot also decreased as the duration of cold storage increased; this parameter was also significantly affected by the storage time in all clones except Q10, Q3, and VA11-Bs (Table 5). After 12 months, cultures generally showed signs of necrosis, in both the shoot apex of explants and the upper leaves, and slight signs of etiolation because the material was stored in low light conditions. (Figure 2A). Nonetheless, these cold-stored shoot cultures were able to recover normal growth after 4 weeks of culture on standard conditions (Figure 2B).

*3.2. Somatic Embryogenesis*

Despite the large number of auxin treatments (IBA, NAA, or IAA), auxin exposure times (2, 4, or 8 weeks), and explant types (leaves, shoot tips, or nodes) evaluated in clones VA11-C and Bs, SE was only successfully achieved in VA11-C (1%) when apex explants were exposed to induction medium without plant growth regulators. In this medium, embryogenesis seemed to develop directly on the totally necrotic explants without prior formation of a callus. However, when the induction medium included PGRs, a large compact callus began to develop from which numerous roots emerged after transfer to the expression medium without regulators, although no somatic embryos were formed. When shoot apex and nodal explants of VA5 and PL-T2 clones were cultured on PGR-free medium, somatic embryos were only formed in both explants of VA5 but at a low

frequency (1–2%). Generally, only one somatic embryo/nodular structure was formed per embryogenic explant (Figure 1F), which are maintained according to [26].

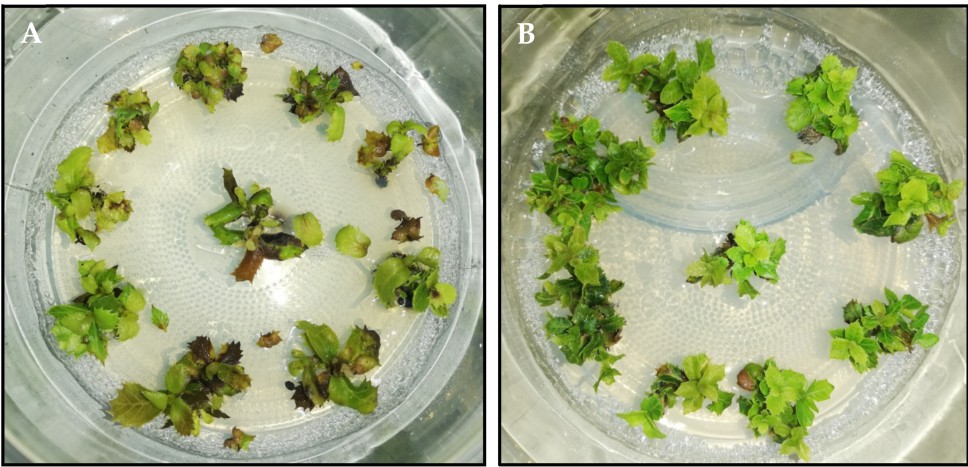

**Figure 2.** Medium-term conservation of axillary shoot cultures of holm oak by cold storage. (**A**) Appearance of shoot cultures of clone VA11-Bs after storage for 12 months at 4 °C under dim light conditions. (**B**) Axillary shoot proliferation of VA11-Bs clone after 12 months of cold storage and 4 weeks on proliferation medium under standard conditions. Jar diameter: 90 mm.

### 3.3. Dual Cultures

After 3 days of culture (Figure 3), significant ($p = 0.009$) inhibition of *Pc* mycelium growth was evident when confronted in vitro with shoots of the tolerant PL-T2 genotype than when confronted with the other three genotypes (mean of $1.34 \pm 0.10$ cm front to a mean of $1.47 \pm 0.10$ cm, Figure 4). However, this inhibition was not observed after 6 days, nor after further days of culture (data not shown). Note that mycelium growth was slightly lower in both, PL-T2 and VA5, tolerant genotypes after 6 days of culture (Figure 4).

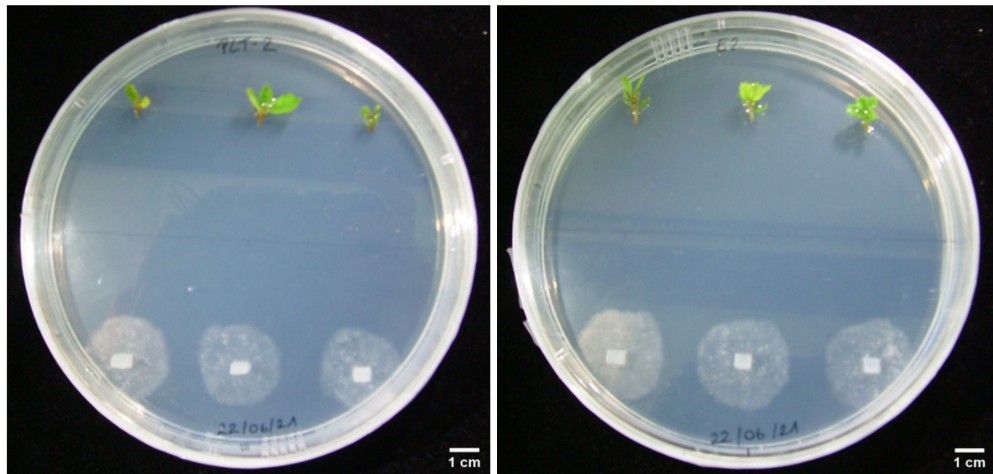

**Figure 3.** Growth of *P. cinnamomi* mycelia after 3 days of dual culture, when confronted with shoots of a *Q. ilex* tolerant (PL-T2, **left**) and a control (E2, **right**) genotypes.

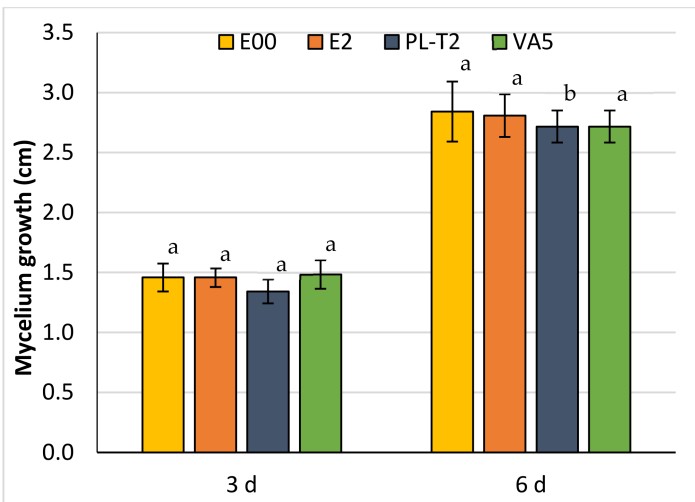

**Figure 4.** *P. cinnamomi* mycelium growth after 3 and 6 days in dual cultures with shoots of *Q. ilex* control (E00, E2) and tolerant (PL-T2, VA5) genotypes. Data are mean ± SD of 4 replicates containing 3 shoots each. For each time, values with the same letter did not differ significantly according to Kruskal–Wallis ANOVA tests.

In the dual cultures performed with leaf explants (Figure 5), *Pc* mycelium growth was initially (after 2 days) slightly inhibited when challenged against the tolerant PL-T2 and VA5 genotypes, although differences were not significant ($p = 0.075$). The genotypic differences in *Pc* growth inhibition persisted after 4 and 7 days ($p = 0.024$ and $p = 0.006$, respectively), particularly for the VA5 genotype (Figure 6). In contrast, after 4 days, mycelium growth rates towards PL-T2 plant material did not differ significantly from those towards E00 and E2 genotypes used as the control.

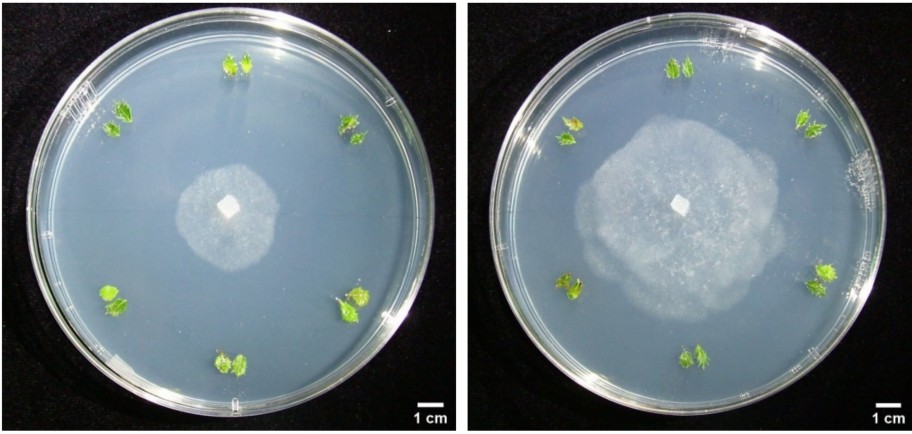

**Figure 5.** Growth of *P. cinnamomi* mycelia after 4 days (**left**) and after 7 days (**right**), towards leaves of the tolerant VA5 (up), and the E2 (bottom left) and E00 (bottom right) control genotypes.

### 3.4. Total Phenols Content of In Vitro Grown Holm Oak Shoots

Basal content in total phenols of in vitro grown shoots from the *Q. ilex* tolerant genotype VA5 were significantly ($p = 0.001$) higher than those contents determined in shoots from the other three genotypes ($31.7 \pm 2.3$ front to a mean of $24.6 \pm 2.9$ μg/g DW). It is worthy to note that PL-T2 had the lowest basal phenols content ($21.2 \pm 2.7$ μg/g DW), while presented the highest levels when confronted with *P. cinnamomi* ($43.3 \pm 4.5$ μg/g DW). This high phenols content contrasted ($p = 0.013$) to those determined in E2 and E00 shoots sampled in infected plates ($33.0 \pm 5.3$ and $34.3 \pm 2.2$ μg/g DW, respectively) that did not differ from levels determined in VA5 shoots confronted with *Pc* ($35.1 \pm 3.1$ μg/g DW).

These results indicate that the response to *Pc* infection of the in vitro growing holm oak shoots was substantially dependent on the genotype, since the phenols content did not significantly increase in the E2 and VA5 genotypes while it doubled in PL-T2 plant material (Figure 7).

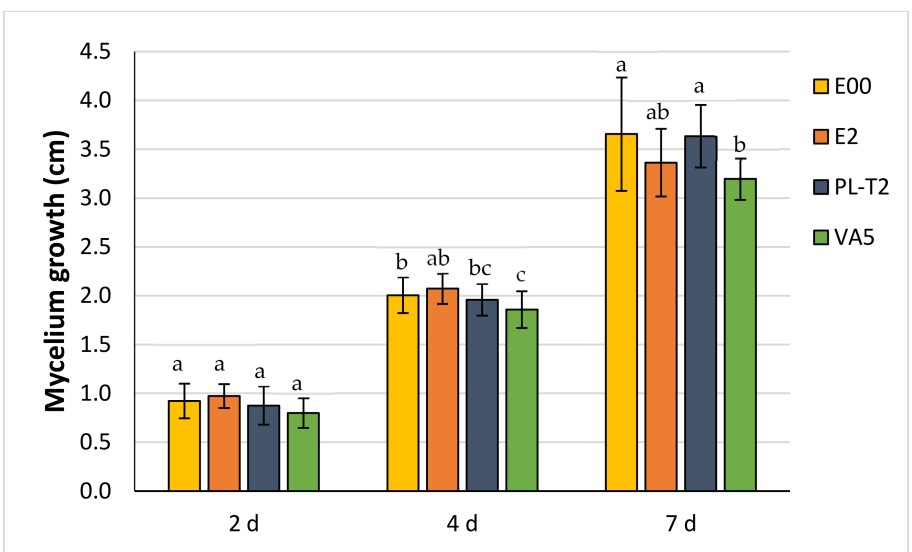

**Figure 6.** *P. cinnamomi* mycelium growth after 2, 4, and 7 days towards leaves excised from in vitro grown shoots of five *Q. ilex* genotypes. Dual cultures were performed in Petri dishes combining explants from a control (E00, E2) and a tolerant (PL-T2, VA5) genotype. Data are mean ± SD of 4 replicates containing 3 pairs of leaves each. For each time, values with the same letter did not differ significantly according to Kruskal–Wallis ANOVA tests.

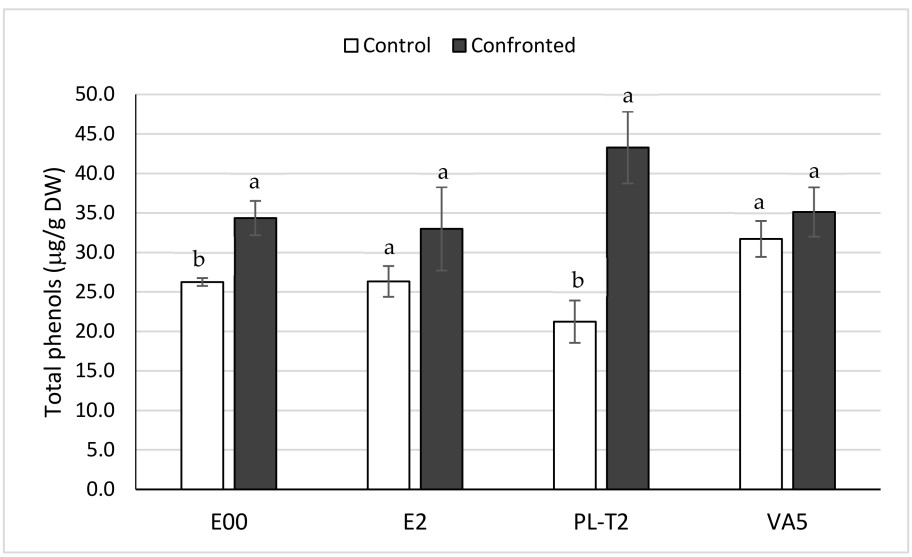

**Figure 7.** Total phenols content of in vitro grown *Q. ilex* shoots from control (E00, E2) and tolerant (PL-T2, VA5) genotypes, after 17 days in uninfected (white bars) and infected (dark bars) plates of dual cultures challenged with *P. cinnamomi*. Data are mean ±SD of two replicated samples analyzed twice. For each genotype, values with the same letter did not differ according to the Tukey-b test.

### 3.5. Gene Expression in Leaves of In Vitro Grown Holm Oak Shoots

Real-time amplification products obtained from *Q. ilex* cDNA were consistent with those of references, and the primer pair efficiency for the five analyzed genes was higher than 94%. The pattern of expression in phenolics biosynthesis-related genes varied among

genotypes before and, specially, after being challenged to *P. cinnamomi* in dual cultures (Figure 8). The basal expression of the four genes was significantly different among the three genotypes studied ($p = 0.023$, $p < 0.001$, $p = 0.027$, and $p = 0.024$, for *CS*, *PAL*, *CAD*1 and *ChS*, respectively), since the leaves of PL-T2 shoots showed higher levels of expression than the other two genotypes (Figure 8). Challenging shoots with the oomycete displayed an *CS*, *PAL*, and *ChS* overexpression profile in both tolerant PL-T2 and VA5 genotypes as compared to E00, while the opposite pattern was observed for the *CAD*1 gene, which was overexpressed in E00 and PL-T2 while it was downregulated in VA5 (Figure 8).

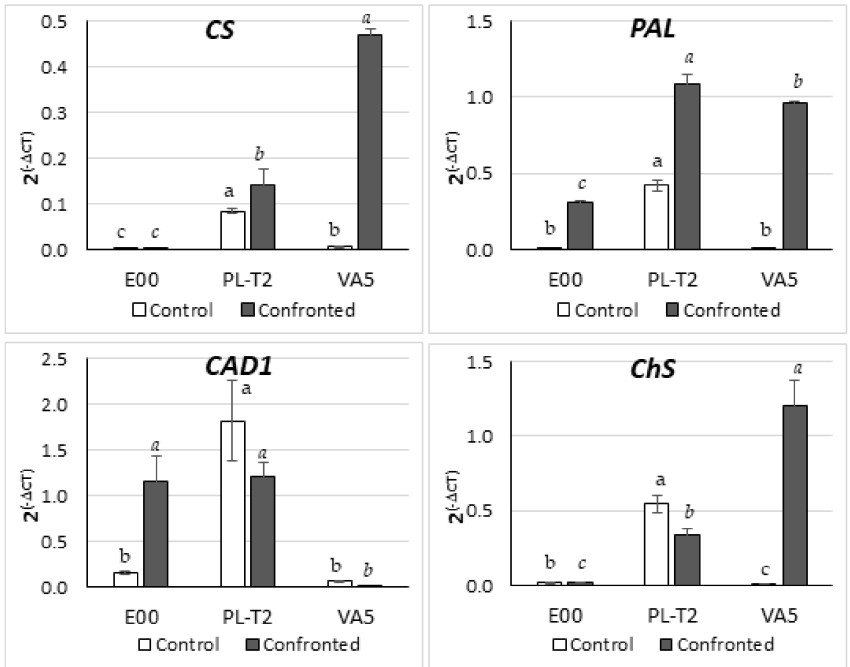

**Figure 8.** Expression of chorismate synthase (*CS*), phenylalanineammonia-lyase (*PAL*), NADPH-dependent cinnamyl alcohol dehydrogenase (*CAD*1), and chalcone synthase (*ChS*) genes in in vitro shoots *Q. ilex* of genotypes E00, PL-T2, and VA5. Data are mean ± SD of amplifications performed in triplicates. Within each gene and treatment, values with the same letter did not differ significantly according to ANOVA tests.

## 4. Discussion

Conventional breeding programs with forest species are time consuming, mainly because of the long lifespan of trees, and biotechnology can be a very valuable tool for generating disease-tolerant genotypes [18,47,48]. Propagation of tolerant genotypes (also called "escape" genotypes) detected by screening natural genetic variation has been suggested as a possible alternative to conventional methods, in order to produce tolerant genotypes in a relatively short time [49]. In the last few years, several researchers have reported that genotypes tolerant to important diseases have been identified in nature. For example, elm trees tolerant to Dutch elm disease have been found across North America and Europe [50–52]. Additionally, genetic variation in resistance to ash dieback in *Fraxinus excelsior* has been detected in several European countries [53,54]. However, both host and pathogens can display different types of behavior, and infected trees may therefore appear to be asymptomatic, e.g., when no attack occurs, because the parasite and host life cycles do not coincide. Thus, since "escape" trees cannot be inoculated in the field to test their tolerance, they must be tested under controlled laboratory conditions, which imply that they must be cloned for posterior confirmation of their tolerance through infection assays or detection of tolerance markers.

Based on this approach, in the present study, holm oak trees displaying no symptoms were identified in areas where neighboring trees exhibited severe symptoms. These trees

were subsequently propagated by axillary budding. Several studies have pointed out the recalcitrant character of many species of the *Fagaceae* family to clonal propagation, especially when the starting material is of adult origin ([17] and references therein). Nonetheless, four adult genotypes of holm oak selected for their tolerance to *P. cinnamomi* at critical disease spots were established in vitro. Our findings corroborate that in vitro establishment of axillary shoot cultures derived from flushed shoots removed from branch segments collected from adult trees is possible [55,56]. This method enables growth of dormant (quiescent) buds and often leads to the development of shoots that show some features of rejuvenation, which facilities in vitro establishment [57]. Secondly, in many hardwood species, in vitro establishment is more successful with material taken from the basal part of the trunk than with crown material [17,56,57]. However, no significant differences were observed in the establishment of either type of material in holm oak genotype VA11. This finding is further proof of the extreme recalcitrance of this plant species to vegetative propagation, as has also been reported regarding the in vitro establishment of juvenile plants [30]. Important differences in the success of in vitro establishment were observed between the trees from the two geographical regions considered, since trees from the Mediterranean region yielded better results than those from the southwestern region. These differences can probably be attributed to differences between genotypes or to the different physiological states of the trees related to the disease level. In Plasencia, the location sampled in south-west Spain, *Q. ilex* trees grow in dehesas strongly affected by oak decline, and although they survive the disease, they may be weakened, which will affect the success of in vitro propagation.

Rooting is a crucial step in the success of micropropagation by axillary budding [58]. However, holm oak is considered a difficult-to-root species [59], and the rooting rates reported for this species are low, even in material of juvenile origin [30]. Here, rooting was achieved in the three tested clones, but acceptable rooting values were obtained only in genotype VA5. Significant differences in the rooting capacity of clones established from basal (Bs clones) and crown (C clones) material from the same tree have been reported in chestnut and pedunculate oak, with Bs clones exhibiting a more juvenile character and therefore high rooting capacity than C clones [56]. However, this trend has not been observed in holm oak and we observed a small difference in rooting frequencies between VA11-Bs (12%) and VA11-C (8%).

Regarding the rooting treatment, it was successfully accomplished only in medium with 3 mg L$^{-1}$ IBA plus 0.1 mg L$^{-1}$ NAA. This combination was also the most suitable for shoots derived from adult holm oak trees selected for fruit production [21]. By contrast, for rooting axillary shoot cultures established from juvenile holm oak plants [30], the best results were obtained with IBA 25 mg L$^{-1}$. In the present report, the last treatment was only effective with genotype VA5, and higher values were obtained than in rooting medium supplemented with 3 mg L$^{-1}$ IBA plus 0.1 mg L$^{-1}$ NAA. Our results indicate that in holm oak, the most suitable treatment for rooting must be selected for each genotype. Similarly, the important effect of genotype has also been mentioned in relation to the micropropagation of other hardwood species, such as *Morus alba* [60], American oaks [61], *Arbutus unedo* [58], *Alnus glutinosa* [62], or *Q. lusitanica* [63]. Phoroglucinol usually combined with an auxin has been used to stimulate the rooting on several species [64]; however, on shoot rooting of holm oak, this compound has a detrimental effect, inhibiting rooting.

Slow growth is applied to the conservation of in vitro shoot cultures with the aim of reducing the frequency of periodic subculture, without affecting the viability and regrowth of shoot cultures when they are returned to standard conditions [65]. Cold storage is the most common strategy used to reduce shoot growth, since it is a simple and economical procedure that only requires a cold chamber or a fridge [29]. In the present study, cold storage of shoot cultures derived from adult holm oak trees enabled conservation of six of out seven tested clones, with high recovery rates (68.6–100%) after 12 months. Similar values to those obtained in the present study have been reported for other woody species. For example, in chestnut, survival rates of almost 100% were achieved after 12 months at

4 °C [66]. Likewise, in olive shoots stored in the cold for 8 months, correct growth was regained in 80–90% of the shoots [67]. In black alder, axillary shoot cultures can be stored in the cold for up to 18 months, with survival rates above 75% [68]. For good results to be obtained, careful selection of the initial explant to be stored in cold was important in holm oak, and clumps of two or three actively growing shoots were used.

Somatic embryogenesis was induced in nodes and shoot tips excised from in vitro shoot cultures derived from two adult trees. Induction of somatic embryogenesis in this type of explant provides advantages over the use of floral tissue, as it is independent of climate and seasonal growth, and therefore explants can be produced throughout the year [69]. Our group has demonstrated the feasibility of using explants derived from axillary shoot cultures to induce SE in adult trees of different oak species [70–73] and eucalypts [74]. The type and physiological stage of the initial explant are critical factors for SE induction [22]. In holm oak, apex and nodal explants are the most suitable types of explants for inducing somatic embryos. Likewise, in *Eucalyptus*, higher induction rates were obtained in apex explants than in leaves in all three genotypes evaluated [74]. One possible explanation for this finding is that shoot apices and nodal explants contain meristematic cells that can be precursors of pluri- or toti-potent stem cells, facilitating the induction of SE [75]. The type and concentration of auxin and the exposure time are the parameters most commonly evaluated in terms of the composition of induction media [76]. High concentrations of auxin are usually used to initiate SE in non-zygotic explants, often in combination with low cytokinin concentrations [22]. However, the concentration of auxin used to induce SE seems to be related to the degree of cell dedifferentiation in the initial explant. When the initial explant possesses predetermined embryogenic cells, as may be the case in immature zygotic embryos, somatic embryos are often induced by the use of a high cytokinin-to-low auxin ratio or PGR-free medium. In the present case, culturing apex explants on medium without PGRs produced better results than using high auxin concentrations. This may indicate a low level of cell differentiation in apex and nodal explants from holm oak shoots and that the stress induced by the excision process may be sufficient to redirect the embryogenic pathway. The induction rates obtained in the present study, although low, are similar those reported for zygotic embryos [23], floral tissues [24,25], and shoot explants isolated from in vitro shoots established from juvenile plants [30].

Several experiments were undertaken to corroborate in vitro the tolerance to *Pc* of the generated clones of the VA5 and PL-T2 holm oak genotypes as compared to those from E00 and E2 genotypes that, although its response to *Pc* had not been characterized so far, grew in unaffected areas and were considered as controls. First, to study whether dual culture assays could be of use to determine the in vitro tolerance, isolated shoots and leaves were challenged against actively growing *Pc* mycelium. Shoots from PL-T2 and leaves from VA5 inhibited the growth of this mycelium (Figures 4 and 5), but the inhibition did not persist for more than 10 days. Other authors, however, successfully reported this technique to test in vitro the virulence of different fungus species against *Pinus sylvestris* [77]. Our results indicate that parameters required to establish the in vitro dual cultures, including the explant used, need to be optimized, since in our conditions, shoots were more suitable for the PL-T2 genotype whereas leaves were for the VA5 genotype.

Besides their role in lignin synthesis and wood formation, phenolic compounds are involved in plant defense responses. For example, a combination of enhanced constitutive and inducible phenolic-based mechanisms has been found to lower the susceptibility of Scots pine to the fungus *Heterobasidion annosum* [78]. The amount and the type of phenylpropanoid-derived compounds vary among the different plant species and also depend on the stress sensitivity of plants and the biosynthetic ability of the tissues [79]. In *Quercus rubra*, studies on phenolic compound contents have been focused on tannins variation during leaf senescence [80]. In our work, the basal level of total phenolics compounds depended on the genotype, being lower or higher in the tolerant PL-T2 and VA5 genotypes, respectively, and middle in the two controls (Figure 7). Differences among holm

oak provenances in response to several stresses at the protein level have been reported [81]. Corroborating the already reported role of the phenylpropanoid pathway in plants' defense mechanisms, higher levels of phenolic compounds were found in the in vitro growing shoots from tolerant genotypes after being challenged against *Pc* mycelium, although differences were significant only for the PL-T2 clonal progenies that reached 43 µg GAE/g DW, whereas those from E00 and E2 genotypes did not exceed 35 µg GAE/g DW. When investigating the significance of the secondary metabolism for the susceptible and resistant interaction between *Eucalyptus* saplings and *P. cinnamomi,* it was demonstrated that inoculation of roots with *Pc* induced the activity of *PAL* and the concentration of phenolic compounds in root segments of the resistant species *Eucalyptus calophylla* but not in the susceptible *Eucalyptus marginata* [82].

Several molecular approaches have been applied to study the *Q. suber–P. cinnamomi* pathosystem in order to disclose pathogenesis-related genes (see [37,83]), a recent and extensive review [84] focused on omics analyses of the stress response in *Quercus* spp is also available. The main *Q. suber* genes characterized in this pathosystem were those coding for phenylalanine ammonia lyase (*PAL*), cinnamyl alcohol dehydrogenase2 (*CAD*2), CAD1 (*CAD*1), NBS-LRR resistance protein *(RPc)*, RelA/SpoT protein *(RSH)*, disulphide isomerase *(PDI)*, and a cationic peroxidase *(POX1)* [37]. To study the defense response in our regenerated shoots, we determined the expression of four genes involved in phenylpropanoid biosynthesis before and after *Pc* infection. First, the chorismate synthase (*CS*) catalyzes the formation of chorismate, the final product of the shikimate pathway and precursor of aromatic aminoacids, and some hormones, such as salicic acid [85]. This gene was upregulated in both tolerant genotypes, PL-T2 and VA5, after challenging with *Pc*, indicating activation of the phenylpropanoids pathway. Second, phenylalanine ammonia-lyase (*PAL*) catalyzes the first committed step of the core pathway of general phenylpropanoid metabolism [86]. The expression of this gene increases markedly in many plant–pathogen interactions, in response to microbial or endogenous elicitors, and is therefore considered a chemical marker of induced resistance in many plants [87]. In our work, the *PAL* gene was upregulated in the three analyzed genotypes after *Pc* challenge, but its expression was especially higher in the two tolerant genotypes (Figure 8), which corroborates its oomycete tolerance.

For a tolerant interaction, it is proposed that effector molecules like aldehyde aromatic compounds, similar to the eutypine toxin, released by *P. cinnamomi* during early infection, can be reduced and inactivated to alcohols by a *Q. suber* cinnamyl alcohol dehydrogenase 1 (*CAD*1) [88]. The tolerant genotype PL-T2 displayed a highly upregulated phenotype for this gene before being in contact with *Pc* (Figure 8), indicating the existence of a degree of a defense barrier before stress. Finally, chalcone synthase is a key enzyme of the flavonoid/isoflavonoid biosynthesis pathway that is part of the plant developmental program, and that is also induced in plants under stress conditions [89]. For example, a high flavonoid content has been reported in adult holm oak after drought stress [84]. In our experiment, the *CHS* gene was upregulated in in vitro clonal progenies of tolerant holm oak genotypes before (PL-T2) or after (VA5) being infected with *Pc*.

All these results suggest that the two tolerant genotypes selected in the field maintain tolerance profiles after being cloned in vitro. Tolerance mechanisms are different in the two genotypes since PL-T2 from the south-west provenance displayed a phenotype with a basal low phenolic content and upregulation of the *CS*, *PAL*, *CAD*1, and *CHS* genes before infection, while the VA5 genotype from the Mediterranean area had a high basal phenol content and downregulation of the four genes before infection. In both genotypes, challenge against *Pc* produced significant increases in *CS*, *PAL*, and *CHS* gene expression as compared to a control E00 genotype.

## 5. Conclusions

The protocols defined in the present report show the great possibilities of the application of micropropagation procedures for propagation, conservation, and tolerance

evaluation of selected *Q. ilex* genotypes in disease hotspots. However, despite the achievements, axillary budding and somatic embryogenesis protocols must be further improved in relation to increasing the rooting rates and induction embryogenic frequencies. Cold storage of in vitro shoot cultures of adult holm oak trees provides a simple and efficient protocol for conservation of the genetic diversity of this species or while their tolerance to Pc is characterized. Dual cultures can be a simple procedure for testing holm oak tolerance to *Pc* in vitro, although the protocol needs to be optimized to obtain consistent results for individuals from different provenances. Higher levels in total phenols and of the expression of genes regulating the phenylpropanoids biosynthesis pathway were associated with holm oak trees displaying a field-tolerance phenotype.

**Supplementary Materials:** The following are available online at https://www.mdpi.com/article/10.3390/f12121634/s1, Table S1: Details of *Quercus ilex* trees selected for their tolerance to *P. cinnamomi* in two different areas of Spain. Table S2: Effects of the exposure time (24 h versus 48 h) to 25 mg L$^{-1}$ IBA and the genotype/ontogenic origin on the rooting of shoots of the VA5 and VA11-Bs and C clones.

**Author Contributions:** Conceptualization, methodology and investigation of micropropagation objectives, M.T.M. and E.C.; conceptualization, methodology and investigation of in vitro shoot cultures characterization, I.A., E.S., M.d.C.G.-M. and M.A.P.-O.; writing—original draft preparation, M.T.M., I.A., E.S. and E.C. All authors have read and agreed to the published version of the manuscript.

**Funding:** This investigation was partly supported by the projects AGL2016-76143-C4-4-R, AGL2016-76143-C4-1-R, PID2020-112627RB-C33 and PID2020-112627RB-C31. We also acknowledge support of the publication fee by the CSIC Open Access Publication Support Initiative through its Unit of Information Resources for Research (URICI).

**Acknowledgments:** We thank M. Toribio and M.M. Ruiz Galea (IMIDRA) for kindly providing the branch segments collected on holm oak trees located on Plasencia (Cáceres, Spain). We also thank P. Abad (UPV) for providing the *Phytophthora cinnamomi* strain.

**Conflicts of Interest:** The authors declare no conflict of interest.

## Abbreviations

| | |
|---|---|
| BA | benzyladenine |
| Bs | basal part of the tree |
| C | crown of the tree |
| CAD | NADPH-dependent cinnamyl alcohol dehydrogenase |
| ChS | chalcone synthase |
| CS | chorismate synthase |
| EF | elongation factor |
| GD | Gresshoff and Doy medium |
| IAA | indole-3-acetic acid |
| IBA | indol-3-butyric acid |
| MS | Murashige and Skoog medium |
| NAA | α-naphthalene *acetic* acid |
| NSs | nodular embryogenic structures |
| PAL | phenylalanineammonia-lyase |
| Pc | *Phytophthora cinnamomic* |
| PDA | Potato Dextrose Agar medium |
| PGRs | plant growth regulators |
| PPA | Plant Propagation Agar |
| qPCR | quantitative PCR |
| SE | somatic embryogenesis |
| SH | Schenk and Hildebrandt |
| STS | silver thiosulphate |
| WPM | Woody Plant Medium |

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
