# Peer review of "Micropropagation, Characterization, and Conservation of Phytophthora cinnamomi-Tolerant Holm Oak Mature Trees"

_forests, doi:10.3390/f12121634_

Round 1

Reviewer 1 Report

This is a very interesting paper and I have done only a few comments. There are no problems with language or terminology. The results were presented very well, and only some issues would be better to make a little bit improved. I am sure that all points mentioned in ms will be able to make better easily.

Introduction: Basically, I`d suggest reducing the amount of this part and simplifying the type of information

For instance,

Line 45-50: my suggestion is to reduce this paragraph excluding long description of Dehesas.

Line 55-67 – the same, too long description of oak decline. It sounds like repetition and a long explanation for the importance of oak decline and Dehesas.

M&M

Line 135 – Quercus ilex would be better

Line 152 – add temperature and light/darkness for culturing

Line 156-157 Have you used the sterilisation for perlite, plastic tray and so on? Please clarify

Line 265 -266 – days instead d

Line 380-391 - Table 3 should be formatted

Author Response

See enclosed file.

Reviewer 2 Report

This is an interesting paper about the strategies propagate plants to better adapt to unfavourable environmental conditions. 
I have identified a few mistakes that are necessary to correct!

Page 3 line 137 – 139: - It would be good to specify the GPS coordinates of the localities.

Page 3 line 144 – 145: - replace Phytophthora cinnamomi with P. cinnamomi

Page 3 line 149: - replace Phytophthora cinnamomi with P. cinnamomi

Page 6 line 258: - replace Quercus ilex with Q. ilex

Page 6 line 268: - replace Quercus ilex with Q. ilex

Page 7 line 306: - replace Phytophthora cinnamomi with P. cinnamomi

Page 14 line 484: - replace Phytophthora cinnamomi with P. cinnamomi

Page 14 line 485: - replace Quercus ilex with Q. ilex

Page 14 line 488: - replace Quercus ilex with Q. ilex

Page 14 line 487: - replace Phytophthora cinnamomi with P. cinnamomi

Page 15 line 511: - replace Phytophthora cinnamomi with P. cinnamomi

Page 15 line 515: - replace Phytophthora cinnamomi with P. cinnamomi

Page 15 line 516: - replace Quercus ilex with Q. ilex

Page 15 line 521: - replace Quercus ilex with Q. ilex

Page 15 line 525: - replace Phytophthora cinnamomi with P. cinnamomi

Page 16 line 534: - replace Quercus ilex with Q. ilex

Page 16 line 536: - replace Phytophthora cinnamomi with P. cinnamomi

Page 16 line 540: - replace Quercus ilex with Q. ilex

Page 16 line 543: - replace Phytophthora cinnamomi with P. cinnamomi

Page 17 line 554: - replace Quercus ilex with Q. ilex

Page 17 line 575: - ‘neighboring’ not ‘neighbouring’  

Page 19 line 692: - It is mentioned for the first time in the paper E. calophylla and E. marginate. Therefore, full names should be given ‘Eucalyptus calophylla and Eucalyptus marginata’

Page 20 line 703: - replace ‘catalizes’ with ‘catalyzes’

Author Response

See enclosed file.
